# CaMKIIα Expressing Neurons to Report Activity-Related Endogenous Hypoxia upon Motor-Cognitive Challenge

**DOI:** 10.3390/ijms22063164

**Published:** 2021-03-20

**Authors:** Umer Javed Butt, Imam Hassouna, Laura Fernandez Garcia-Agudo, Agnes A. Steixner-Kumar, Constanze Depp, Nadine Barnkothe, Matthias R. Zillmann, Anja Ronnenberg, Viktoria Bonet, Sandra Goebbels, Klaus-Armin Nave, Hannelore Ehrenreich

**Affiliations:** 1Clinical Neuroscience, Max Planck Institute of Experimental Medicine, 37075 Göttingen, Germany; butt@em.mpg.de (U.J.B.); hassouna@em.mpg.de (I.H.); agudo@em.mpg.de (L.F.G.-A.); steixner@em.mpg.de (A.A.S.-K.); barnkothe@em.mpg.de (N.B.); zillmann@em.mpg.de (M.R.Z.); ronnenberg@em.mpg.de (A.R.); bonet@em.mpg.de (V.B.); 2Department of Neurogenetics, Max Planck Institute of Experimental Medicine, 37075 Göttingen, Germany; depp@em.mpg.de (C.D.); sgoebbels@em.mpg.de (S.G.); nave@em.mpg.de (K.-A.N.)

**Keywords:** physiological hypoxia, complex running wheel, hippocampus, scRNA-seq, light-sheet microscopy, Hif-1α, neuronal differentiation, brain maturation, neuron culture, immature neurons

## Abstract

We previously introduced the brain erythropoietin (EPO) circle as a model to explain the adaptive ‘brain hardware upgrade’ and enhanced performance. In this fundamental circle, brain cells, challenged by motor-cognitive tasks, experience functional hypoxia, triggering the expression of EPO among other genes. We attested hypoxic cells by a transgenic reporter approach under the ubiquitous CAG promoter, with Hif-1α oxygen-dependent degradation-domain (ODD) fused to CreERT2-recombinase. To specifically focus on the functional hypoxia of excitatory pyramidal neurons, here, we generated CaMKIIα-CreERT2-ODD::R26R-tdTomato mice. Behavioral challenges, light-sheet microscopy, immunohistochemistry, single-cell mRNA-seq, and neuronal cultures under normoxia or hypoxia served to portray these mice. Upon complex running wheel performance as the motor-cognitive task, a distinct increase in functional hypoxic neurons was assessed immunohistochemically and confirmed three-dimensionally. In contrast, fear conditioning as hippocampal stimulus was likely too short-lived to provoke neuronal hypoxia. Transcriptome data of hippocampus under normoxia versus inspiratory hypoxia revealed increases in CA1 CaMKIIα-neurons with an immature signature, characterized by the expression of *Dcx, Tbr1, CaMKII*α, *Tle4*, and *Zbtb20*, and consistent with accelerated differentiation. The hypoxia reporter response was reproduced in vitro upon neuronal maturation. To conclude, task-associated activity triggers neuronal functional hypoxia as a local and brain-wide reaction mediating adaptive neuroplasticity. Hypoxia-induced genes such as EPO drive neuronal differentiation, brain maturation, and improved performance.

## 1. Introduction

Erythropoietin (EPO) is a potent hypoxia-inducible growth factor, originally named after its role in hematopoiesis, but later described to have neuroprotective functions in the nervous system [1,2,3]. ‘Brain doping’, in the sense of substantially improved maximal exercise performance without altered red blood cell production, has been suspected for a long time to be the main operative force rather than blood doping alone [4]. This was elegantly demonstrated a few years ago in a transgenic mouse line, which constitutively overexpresses human erythropoietin (EPO) exclusively in the brain [5]. However, the mechanisms explaining these dramatic erythropoiesis-independent effects and, in particular, the underlying physiology have remained widely obscure. We previously introduced the brain EPO circle as a working model explaining the adaptive ‘brain hardware upgrade’ and enhanced performance. In this fundamental regulatory circle, brain cells, challenged by motor-cognitive tasks, experience functional hypoxia, triggering the local expression of EPO, among other genes [6]. We confirmed functional hypoxic cells by a transgenic reporter approach under the ubiquitous CAG promoter, with the Hif-1α oxygen-dependent degradation-domain (ODD) fused to CreERT2-recombinase [7].

The brain EPO circle matches perfectly with well-established, but mechanistically unexplored, experimental and clinical observations that physical activity and cognitive challenge induce comprehensive brain activation, thereby improving global brain function and increasing brain dimensions [8,9,10]. We showed that physiological, endogenous hypoxia is likely a respective lead mechanism, regulating neuroplasticity via assimilated gene expression in neurons of the behaving brain [7]. Importantly, targeted deletion of the EPO receptor from pyramidal neurons eliminated the observed improvement in performance [6].

These observations placed pyramidal neurons in the center of our interest and stimulated us to generate a new mouse line using a pyramidal neuronal promoter. The CaMKIIα-CreERT2-ODD::R26R-tdTomato mice presented here allowed us to specifically concentrate on the functional hypoxia of the excitatory pyramidal neurons using immunohistochemistry including light-sheet microscopy (LSM) for three-dimensional views. Additionally, hippocampal single-cell mRNA-seq transcriptome data under normoxia versus inspiratory hypoxia, focusing on CaMKIIα expressing neurons, revealed the first hints of accelerated neurodifferentiation, pointing to enhanced neuroplasticity and brain maturation upon hypoxia.

## 2. Results and Discussion

### 2.1. Generation of a New Mouse Line to Report Pyramidal Neuronal Hypoxia

In our previous work [7], we used a transgenic mouse line, driven by the universal CAG promoter (CAG-CreERT2-ODD::R26R-tdTomato, slightly modified from [11]).This reporter line allowed permanent labelling of all cells undergoing transient hypoxia by the expression of a fusion protein, comprised of Hif-1α ODD and tamoxifen-inducible CreERT2. Immunohistochemical analyses identified mainly neurons as reporter-labeled cells, but also certain (much lower) percentages of astrocytes, oligodendrocytes, endothelial cells, and pericytes. Hence, to specifically focus on functional hypoxia of activity-challenged pyramidal neurons, we created CaMKIIα-CreERT2-ODD::R26R-tdTomato mice (Figure 1a).

To characterize these mice, we started out with our previous ‘gold standard’ of positive control derived from our experience with the CAG-line [7]. We compared mice exposed to intermittent inspiratory hypoxia (6% O_2_ for 6 h daily, applied over five days) with mice under inspiratory normoxia (21% O_2_) that were handled identically, including five tamoxifen injections as indicated (Figure 1b). This regimen led to very strong staining of the pyramidal neurons already under normoxia and an extreme labelling under hypoxia, as illustrated in Figure 1c. Using corn oil injections alone as a solvent control (‘no tamoxifen’) to assess the non-induced expression of the transgene (‘leakiness’ of the tdTomato reporter) led to hardly any labelled cells (Figure 1c; Appendix A).

### 2.2. Critical Considerations Regarding Our Model and Proof-of-Concept Testing

The pCaMKIIα-CreERT2-ODD transgene encodes a Cre-fusion protein that is made and immediately inactivated by two independent posttranslational mechanisms, which are both active: (1) The estrogen receptor (ER) domain is chaperoned by heat shock protein Hsp90 and thus the entire fusion protein is maintained inactive in the cytoplasm: Here, no Cre recombination is possible; (2) The ODD domain causes the hydroxylation of the entire fusion protein (by PHD/prolyl-hydroxylase domain proteins, followed by polyubiquitination via the VHL proteins/Von Hippel-Lindau tumor suppressor, leading to proteasomal degradation). Also here, no Cre recombination is possible. Cre recombination only occurs when both inhibitory mechanisms are blocked at the same time, i.e., by the coincidence of tamoxifen (binding to the ER domain and releasing Hsp90) and hypoxia (inactivating PHD and thereby stabilizing CreERT2-ODD). Nuclear Cre activity, even if short and only at a low level, causes the permanent activation of our Rosa26-tdTomato reporter gene by removing a STOP sequence from the DNA upstream of its coding sequence (Figure 1a), rendering the RNA transcript translatable. Consequently, all Cre-recombined cells become stably red labelled (independent of further tamoxifen and/or hypoxia). Thus, we achieved permanent staining of once hypoxic cells.

However, in order to work, a sufficient concentration of Hsp90 as well as of PHD/VHL proteins to match the cellular abundance of CreERT2-ODD are needed. Too much CreERT2-ODD could out-titrate either Hsp90 and/or PHD+VHL proteins and thus result in unspecific Cre activity (‘leakiness’ of the system). Reassuringly, this is very low in our system as described above (Figure 1c; Appendix A).

On the other hand, the abundance of neuronally expressed CreERT2-ODD may be critical for the PHD/VHL system to work. When expressed under the control of strong promoters (both, pCAG used in [7], and pCaMKIIα are strong promoters), the ODD domain may out-titrate the abundance of PHD and/or VHL proteins, whose expression level is regulated to exactly match the endogenous Hif proteins. These operate at a much lower abundance level, are not ‘inducible’ as Hsp90 proteins are, and PHD/VHL out-titration should not cause cellular stress. Therefore, we checked by qPCR whether known target genes of Hif-1α are upregulated under normoxia and without any hypoxia stimulation, simply due to a high-level expression of the ODD domain in transgenic mice. Again reassuringly, the comparative qPCR results for the selected Hif-regulated genes *Vegfa*, *Higdf1a*, and *Epo*, obtained under normoxic conditions from the hippocampal tissue of pCaMKIIα-CreERT2-ODD and pCAG-CreERT2-ODD versus wildtype mice of the same background strain, age, and gender, did not show differences (data not shown).

### 2.3. Evaluation of a Hypoxia/Tamoxifen Dose–Response Curve

Encouraged by the outcome of these necessary controls, we next performed comparative quantifications of tdTomato+ neurons upon tamoxifen injection immediately before hypoxia, applied either once, thrice, or five times (each 6% O_2_ for 6 h), or analogously before normoxia, in cornu ammonis hippocampi (CA1 and CA3), as well as in the dentate gyrus. The quantification results made us select the three-times tamoxifen/hypoxia condition as a suitable titration result, i.e., as the appropriate ‘gold standard’ of tamoxifen injections to be applied in this particular mouse line (Figure 1d,e).

### 2.4. Immunohistochemical Characterization of Hypoxic Neurons in Brain and Primary Hippocampal Neuronal Cultures

Figure 2 provides illustrating immunohistochemical images of functional hypoxia, giving examples of tdTomato-stained excitatory neurons in the hippocampus and its subfields, various cortical areas and striatum/corpus callosum. We note the strong tdTomato labelling of hypoxic pyramidal neuronal cell bodies and the abundant neuropil staining in the hippocampus. In particular, mossy fibers in the vicinity of CA3 (stratum lucidum) and granule cell projections in the dentate gyrus are strongly labelled. The whole dentate gyrus reveals densely packed hypoxic neurons with intense fiber staining.

In primary E17 hippocampal neuronal cultures, immediate addition of (Z)-4-hydroxytamoxifen (2 µM) from day in vitro (DIV) 0 onwards, under continuous hypoxia (1% O_2_) versus normoxia (21% O_2_), revealed tdTomato labelling on DIV16, clearly enhanced under hypoxia. In contrast, no staining was seen on DIV7 (data not shown), indicating that some degree of neuronal maturation is a prerequisite for the activity of the CaMKIIα promoter, and thus for the expression of the CreERT2-ODD fusion protein, leading to permanent labelling of hypoxic cells (Figure 3a–d).

### 2.5. A Challenging Complex Motor-Cognitive Task but Not a Brief Conditioning Stimulus Induces Functional Neuronal Hypoxia in the Hippocampus

Using complex running wheel (CRW) performance as a challenging motor-cognitive task, a distinct increase in functional hypoxic neurons was assessed immunohistochemically in eight-week-old male and female mice (Figure 4a–c). Particularly in the dentate gyrus, tdTomato labeled neurons were highly abundant under functional hypoxia. Similar to what we observed upon inspiratory hypoxia quantifications (Figure 1e), they were rarer in CA1 and much rarer in CA3. As reported and extensively discussed earlier with the CAG-CreERT2-ODD::R26R-tdTomato mice [7], females displayed distinctly stronger labelling as compared to males (overall gender comparison *p* values CA1: 7.5 × 10^−5^, CA3: 0.0034 and dentate gyrus: 0.00035; unpaired Welch’s *t*-test, two-tailed).

We next wondered whether a very brief conditioning stimulus, leading to an extended imprinting of a negative memory (in this case, foot shocks in a defined context, also combined with a tone), would equally result in functional hypoxia of the involved hippocampal neurons. In contrast to CRW, however, contextual fear conditioning as hippocampal stimulus combined with cue memory as primarily amygdala readout was likely too short-lived and ‘weak’ to provoke detectable neuronal hypoxia in a pilot group of female mice (Figure 4d,e).

Lightsheet microscopy allowed a three-dimensional presentation of stained neurons, with fly-through stacks yielding an ostensive dose-response curve from the nearly negative ‘no tamoxifen’ condition (lack of substantial non-induced expression of the transgene) to normoxia and to hypoxia (Appendix A). In fact, CRW-induced functional neuronal hypoxia was so strong that the tdTomato labeling reached the overall level obtained by inspiratory hypoxia, offering to pool and analyze these two hypoxia conditions together (Figure 5a–c). LSM quantifications under hypoxia (both functional and inspiratory) compared to normoxia (sitting controls) showed a clear trend towards an increase but failed statistical significance due to the considerable mouse-to-mouse variation resulting in a relatively large scatter (Figure 5c; Appendix A).

### 2.6. Transcriptome Data Reveal Profound Regulation of CamkIIα Expressing Neurons Upon Hypoxia vs. Normoxia

We next employed our scRNA-seq dataset of mouse hippocampus (from CAG-CreERT2-ODD::R26R-tdTomato mice) collected upon normoxia (21% O_2_) versus inspiratory hypoxia (6% O_2_ for six hours/day) for five consecutive days (compare Figure 1b; GSE162079 [7]), to evaluate the hypoxia regulation of all cells expressing the CaMKIIα-promoter. In the first step, we selected all glutamatergic neuronal populations (including mossy fiber cells), known to express *CaMKIIα* [12,13,14,15] from the hippocampal dataset and compared the expression levels of *CaMKIIα* under normoxic and hypoxic conditions (Figure 6a). Surprisingly, *CaMKIIα* was consistently upregulated under hypoxia in the excitatory neurons from the three discernible hippocampal regions (CA1, CA3, dentate gyrus). Given that CaMKIIα is known to be a crucial mediator of synaptic plasticity, longterm potentiation as well as memory formation and learning [16,17,18,19], this hypoxia-dependent regulation might point to a common pathway that is activated via endogenous functional hypoxia upon neuronal activation and mirrored by exogenous inspiratory hypoxia. In fact, it could very well support our working model of functional hypoxia-induced ‘brain hardware upgrade’.

Importantly, neither the overall expression level of *CaMKIIα* nor the strength of upregulation under hypoxia in the three hippocampal regions were indicative of the amount of tdTomato-labelling observed in immunohistochemistry; e.g., the expression of *CaMKIIα* and its upregulation under hypoxia was strongest in CA3 (*CaMKIIα* expression CA3 versus CA1: *p* = 1.68 × 10^−52^, logFC = 0.25; CA3 versus the dentate gyrus: *p* = 9.96×10^−9^, logFC = 0.16), whereas the number of tdTomato-labelled neurons (IHC) was lower in CA3 when compared to CA1 and the dentate gyrus (Figure 1e and Figure 4c). Overall, baseline expression of *CaMKIIα* was high, with 97%, 99%, and 93% of excitatory neurons expressing this gene under normoxia in CA1, CA3, and the dentate gyrus, respectively. Thus, although a certain amount of hypoxia ‘over-reporting’—driven by the increased activity of the promoter under hypoxia—cannot be entirely excluded, the regional differences in the strength of hypoxic response we observed with immunohistochemical quantification (tdTomato-labelling) likely reflect true differences in the physiological response of neurons from different hippocampal regions to hypoxia. A systematic error affecting regional differences therefore seems highly unlikely.

In order to further explore the transcriptomic hypoxia response of *CaMKIIα*-neurons, we performed differential expression analyses. Figure 6b depicts the top 20 up- and top 20 downregulated genes under hypoxia after clustering cells and genes by k-means clustering. The major regulated gene clusters included genes of the tubulin (cluster 2) and hemoglobin (cluster 5) family as well as ribosomal genes (cluster 4). Clustering of cells revealed three distinct cell populations with cluster 2 and 3 almost exclusively corresponding to either cells derived from mice exposed to normoxia (cluster 2) or hypoxia (cluster 3). In contrast, cluster 1 was composed of cells from both conditions showing a somewhat ‘intermediate’ expression of hypoxia-regulated genes, e.g., see gene clusters 2, 4, and 5. The subpopulation of cells, which was derived from mice kept at normoxic conditions but appeared to have a mild hypoxia transcriptome signature, might represent cells that recently underwent physiological functional hypoxia. Notably, these cells were mainly situated in the dentate gyrus, a hippocampal region that harbors a neurogenic niche known to require hypoxia for its functionality [21].

In addition, the number of (uniquely) hypoxia regulated genes (|logFC| > 0.25, p_adj_ < 0.05) was highest in the dentate gyrus. Down-sampling of cells included into differential testing (*n* = 2296, i.e., equal to the size of CA3 as the smallest region) was performed beforehand to avoid differences in the number of detectable, differentially expressed genes due to differences in statistical power (Figure 6c, left panel). This finding points again to a specific and elevated responsivity to hypoxia in the dentate gyrus as compared to CA. In total, 89 genes were equally regulated under hypoxia in CA1, CA3, and the dentate gyrus. As shown in Figure 6c, right panel, the percentage of upregulated genes amongst all regulated genes in CA1 and the dentate gyrus was ~54%, whereas in CA3 the majority of genes was downregulated (44% upregulated). Cluster Glut4, a very small cluster characterized by a high expression of immature neuronal markers such as *Dcx*, *Tbr1*, and *Tle4,* showed a remarkable downregulation of genes with ~93% of all hypoxia differentially expressed genes being downregulated. This cluster is highly similar to the cluster of immature CA1 neurons responding to EPO already six hours after rhEPO administration that we identified earlier [6]. At this point, the physiological relevance of the strongly dampening gene regulatory response in this immature cell subpopulation remains unclear, but certainly will be investigated in future studies.

Next, we investigated numerical abundance changes in the different excitatory neuron populations of the hippocampus using MiloR [20]. This tool allows a fine-grained analysis of numerical abundance shifts under hypoxia by forming neighborhoods of similar cells within each cell cluster and assessing their numerical abundance in each treatment condition. Figure 6d,e show the numerical shifts in each neighborhood. In CA1 and the dentate gyrus, numerically increased (red) as well as numerically decreased (blue) neighborhoods were detected under hypoxia. In contrast, CA3 neurons appeared to primarily respond to hypoxia with a numerical decrease (blue). Interestingly, in the CA1 neurons, there was a small subpopulation of cells located at the rim of the cluster (red oval in Figure 6d), which were increased in number under hypoxia. Inspection of expression markers indicated that these cells were of immature neuronal identity as shown by the high expression of *Tbr1*, *Zbtb20*, *Tle4*, and *Dcx* (Figure 6f). In addition, *CaMKIIα* expression, shown to be important for neuronal maturation in the hippocampus [22], appeared to be elevated in these cells (Figure 6f). Interestingly, previous literature inconsistently reported both negative as well as positive effects of exogenously applied hypoxia on neuronal development, differentiation, and maturation [23,24,25,26,27,28,29]. Our current findings indicate a positive, stimulating effect on neuronal differentiation, resulting in an increased number of young neurons in CA1. Importantly, this apparent incongruence of existing findings might not only be due to the different modes and severity of hypoxia application, but also due to a previous lack of more distinct inspections of different neuronal precursor and immature neuron populations in the hippocampus, which we provide here.

### 2.7. Synopsis of Our Model: Hypoxia as a Fundamental Driving Force of Demand-Oriented Neuroplasticity and the Consequential ‘Brain Hardware Upgrade’

In an extensive line of research, we have shown over the last two decades that rhEPO treatment improves cognition and increases brain matter in humans and rodents, both healthy and diseased, but the underlying mechanisms have been unknown for a long time. Just recently, we discovered a fundamental regulatory circle, in which brain networks, challenged by cognitive tasks, drift into ‘functional hypoxia’ that drives neurodifferentiation and dendritic spine formation via neuronal EPO synthesis [6]. These findings highlight that EPO is not only a crucial mediator of neurogenesis during embryonic life and brain development [30,31], but also pivotal for adult hippocampal neurodifferentiation and neuroplasticity on demand. The neuroprotective and anti-apoptotic effects of EPO, promoting neuronal survival, together with its neurotrophic properties, certainly contribute to the observed enhanced differentiation of local silent precursors and to their undisturbed maturation.

From the discovery of this regulatory EPO circle, driving challenge-induced brain maturation, we moved on to study the hypoxia response of all brain cells [7], and in the present work specifically of pyramidal neurons, employing our transgenic mouse models pCAG-CreERT2-ODD and pCaMKIIα-CreERT2-ODD, respectively.

In the sense of ‘dosis facit venenum’, strong oxygen deficit or hypoxia in the brain, as seen after cardiac arrest or in stroke, is definitely a serious state of emergency leading to permanent brain damage. Nevertheless, as shown here, functional hypoxia can be an important signal for growth and a driving force of the ‘brain hardware upgrade’.

Hypoxia induces a specific transcriptional program, enabling cells to adapt to lower oxygen levels and/or to inadequate metabolic support. The transcription is to some degree controlled by hypoxia-inducible factors, binding to hypoxia-responsive elements to modulate the expression of many genes, some of which are potent growth factors such as vascular endothelial growth factor (VEGF) or EPO. Newer work, however, has revealed that hypoxia-induced gene transcription is partly Hif-independent [32,33]. Hence, the immunohistochemical findings reported here including LSM are based on Hif mechanisms, and may, therefore, only partly reflect the global cellular hypoxia response. In contrast, our transcriptome data seem to approach the whole hypoxia regulation more fundamentally. Overall, they strongly support the pivotal effects of hypoxia on neuronal differentiation and brain maturation.

Ultimately, moderate hypoxia, at least partly via brain EPO expression, may evolve as a highly promising add-on treatment strategy for neurodevelopmental, neuroinflammatory, and neurodegenerative brain diseases, applicable in the sense of individualized therapeutic approaches to still untreatable conditions.

## 3. Materials and Methods

### 3.1. Generation of the CaMKIIα-CreERT2-ODD-pAa Vector

The genetic strategy for the generation of the CaMKIIα-CreERT2-ODD-pA vector is illustrated in Figure 1a. Briefly, the construct consists of the murine CaMKIIα promoter, active in excitatory neurons, which drives the expression of tamoxifen-inducible Cre-recombinase (Cre-ERT2), fused to the oxygen-dependent degradation domain (ODD) of Hif-1α. The latter responds to varying oxygen concentrations. The pCAG-CreERT2-ODD-pA plasmid was kindly provided by Dr. Hashim A Sadek (USA) [11] and slightly modified [7]. In the first step, the CAG promoter was excised by digesting CAG-CreERT2-ODD-pA with SpeI and EcoRI restriction enzymes. In parallel, the CaMKIIα promoter fragment of 8.5 kb was obtained by digesting the CaMKIIα-HA-cEPOR vector [34,35] with XmaI, NdeI, PacI, and NotI restriction enzymes (New England Biolabs, USA). In a second step, this CaMKIIα promoter fragment was ligated upstream into the CreERT2-ODD vector to yield CaMKIIα-Cre-ERT2-ODD-pA.

### 3.2. CamkIIα-CreERT2-ODD: R26R-tdTomato Transgenic Mice: Generation & Validation

CaMKIIα-Cre-ERT2-ODD-pA was digested with the SfiI and SalI (New England Biolabs, USA) restriction enzymes and the ampicillin selection sequence removed. After gel elusion with the QIAquick gel extraction kit (Qiagen, Venlo, Netherlands), linearized DNA of 11.6 kb was obtained comprising CaMKIIα promoter, inducible Cre-recombinase, and ODD sequence. The linearized DNA was used for pro-nuclear microinjection into fertilized eggs for the production of CaMKIIα-CreERT2-ODD transgenic mice. Litters from the foster mothers were screened for the presence of CaMKIIα and ODD sequence by genotyping as explained below for suitable transgenic founders. CaMKIIα-CreERT2-ODD transgenic mice were maintained on C57BL/6N background (Charles River, MA, USA) and F1 litters carrying the CaMKIIα-CreERT2-ODD transgene were selected and further crossed with C57BL/6N and R26R-tdTomato mice for colony maintenance and to generate the desired heterozygous reporter mice. F2 litters of both genders harboring the CaMKIIα-CreERT2-ODD::R26R-tdTomato transgene were used for the experiments. All transgenic mice showed normal breeding, home cage behavior and life expectancy, and lacked any obvious abnormal phenotype.

### 3.3. Genotyping

Genotyping polymerase chain reaction (PCR kit, Biozym, Oldendorf, Germany) was performed by using the following primers for *CaMKIIα* promoter, forward: 5’-GGTTCTCCGTTT GCACTCAGGA-3’ and reverse: 5′-CCATGAGTGAACGAACCTGG-3 and for ODD forward: 5′-GCTGAAGACACAGAAGCAAA-3′ and reverse: 5′-GTGGGTAGGAGATGGAGATG-3′. In the subsequent generation, CaMKIIα-CreERT2-ODD::R26R-tdTomato transgenic F2 offspring, harboring the tdTomato sequence were screened by the following primers: Primer1: 5′TCAATGGGCGGGGGTCGTT3′; Primer2: 5′CTCTGCTGCCTCCTGGCTTCT3′; Primer3: 5′CGAGGCGGATCACAAGCAATA3′. Complete PCR protocols are available on request.

### 3.4. Mouse Maintenance

All experiments were approved by and conducted in accordance with the regulations of the local Animal Care and Use Committee (Niedersächsisches Landesamt für Verbraucherschutz und Lebensmittelsicherheit, LAVES). CaMKIIα-CreERT2-ODD::R26R-tdTomato mice were kept on the C57BL/6N genetic background, group-housed, maintained in a germ-free and temperature (21 ± 2 °C)-controlled environment on a 12 h light/dark schedule. Adult transgenic mice from F2 generation (seven to eight weeks old) of both genders, harboring the CaMKIIα-CreERT2-ODD::R26R-tdTomato transgene, were used in the experiments unless otherwise stated. Each mouse was single-housed for habituation before the start of the experiment. All mice (normoxia, hypoxia, CRW, ‘no tamoxifen’) were handled in the same way, and euthanised two days after the last tamoxifen/solvent control injection. See the experimental designs in the figures for details.

### 3.5. Tamoxifen Preparation

A stock solution of tamoxifen was prepared by mixing 300 mg tamoxifen in 30 mL of corn oil (both Sigma-Aldrich, Darmstadt, Germany) by sonication at 37 °C and stored at 4 °C.To label the hypoxic cells in normoxia, CRW, or hypoxia mice, a working solution of tamoxifen (100 mg/kg) was freshly warmed up, sonicated for 20 min before each application, and injected intraperitoneally (i.p.) as stated in the experimental designs. The ‘no tamoxifen’ controls received vehicle (corn oil) only.

### 3.6. Hypoxia Application

Exogenous hypoxia was applied to CaMKIIα-CreERT2-ODD::R26R-tdTomato mice in a small hypoxia chamber (60 cm × 38 cm × 20 cm), custom-designed in cooperation with Coy Laboratory Products (Michigan, USA). The hypoxia chamber is connected to an oxygen sensor, a stage controller and a ceiling fan for constant air circulation. In the hypoxia treatment group, CaMKIIα-CreERT2-ODD::R26R-tdTomato mice received a first dose of tamoxifen (100 mg/kg/d) and were then put into the hypoxia chamber. The ambient oxygen concentration (20.9% O_2_) was dropped to 6% O_2_ within 15 min, kept for 6 hand then increased back to 20.9% O_2_. In the normoxia, CRW, and ‘no tamoxifen’ group, mice received tamoxifen injections or corn oil, respectively, and were kept in ambient oxygen concentration (20.9% O_2_).

### 3.7. Complex Running Wheel (CRW) Exposure

CRW is a simple, sensitive, and fully computerized voluntary wheel-running task, which is frequently used to detect motor-cognitive and activity impairments in mice [36,37]. The CRW (TSE Systems, Bad Homburg, Germany) is defined by irregularly spaced bars. The locomotor activity and running performance is computer-controlled via Phenomaster software (TSE Systems, Bad Homburg, Germany). Untrained CaMKIIα-CreERT2-ODD::R26R-tdTomato transgenic mice (eight-week-old; both genders) were given tamoxifen injections every other day (Figure 4a) and had continued free access to CRW for five days. On day five, mice were transferred back to standard cages for two days and later perfused.

### 3.8. Fear Conditioning

Fear conditioning (Med Associates inc, Hertfordshire, United Kingdom) was performed in a single trial (design in Figure 4d) in which, after a 120 s baseline period, mice experienced two paired presentations of a 10 s, 5 kHz, 85 dB tone (conditioned stimulus) and an electrical foot shock for 2 s with an intensity of 0.4 mA (unconditioned stimulus, Context A). The control group (SHAM) consisted of mice exposed to the context only. The contextual memory was assessed 48 h after training (Context A). Trained mice and control mice were observed over 120 s for freezing inside the conditioning chamber. After 4 h, mice were placed in an unfamiliar, new chamber (Context B) and were, after an initial 120 s baseline phase, re-exposed to the tone (CS) to assess cued memory/fear over 120 s.

### 3.9. Hippocampal Neuronal Culture and Image Acquisition

For the neuronal culture, CaMKIIα-ODD-CreERT2::tdTomato fetuses were extracted from the uterus at approximately embryonic day 17 (E17). Hippocampi were dissected from the brain and the meninges were removed before digestion with 0.05% trypsin for 10 min at 37 °C. Subsequent washes were performed in a neuronal culture medium containing 2% B27 (cat. no. 17504044), 1% GlutaMAX (cat. no. 35050087), and 1% penicillin/streptomycin (cat. no. 15140163) in Neurobasal A (cat. no. 10888022, all from Gibco, Thermo Fisher Scientific, Darmstadt, Germany). Single-cell suspension was obtained by careful dissociation of the tissue with a P200 micropipette. Seeding was conducted using poly-D-lysine (PDL) (Sigma-Aldrich, Darmstadt, Germany)-coated glass coverslips in 24-well plates, in densities of 100,000 cells/well. The same day of the preparation, fresh 2 µM (Z)-4-hydroxytamoxifen (H79049, Sigma-Aldrich, Hamburg, Germany) or an equivalent volume of 99.8% pure ethanol (EtOH) (Honeywell, North Carolina, USA) were prepared and added to the cultured wells. The final EtOH concentration was always <0.016%. Cells were incubated at 37 °C in 5% CO_2_, in either normoxia (21% O_2_) or hypoxia (1% O_2_), for seven or 16 days. Then, fixation was performed with 2% acrolein (Sigma-Aldrich, Darmstadt, Germany) and 3% formaldehyde (Merck, Darmstadt, Germany) in PBS. IncuCyte^R^ ZOOM (Sartorius, Ann Arbor, MI, USA) images (1400 × 1400 µm) were taken with a 10× objective using phase contrast and the red fluorescent channel. For the neuronal quantification of tdTomato+ cells, images were processed in IncuCyte^R^ ZOOM software with TopHat parameter (radius = 100 µm, threshold red calibrated unit = 0.5) and edge split sensitivity = −10.

### 3.10. Tissue Preparation and Immunohistochemistry

For tissue harvesting and perfusion, Avertin (Tribromoethanol, Sigma-Aldrich, St Louis, MN, USA, 0.276 mg/g) was used for deep sedation by intraperitoneal injection followed by perfusion with ice cold 0.9% saline solution and 4% formaldehyde (Merck, Darmstadt, Germany). The brains were removed, placed in 4% formaldehyde at 4 °C for 24 h, then cryoprotected in 30% sucrose in phosphate buffered saline (PBS) (Merck, Darmstadt, Germany) at 4 °C for 48 h, finally covered in cryoprotectant (O.C.T.TM Tissue-Tek, Sakura, The Netherlands) and preserved at −80 °C until use. In preparation for immunohistochemistry (IHC), brains were cut into 30-μm-thick coronal sections using a cryostat (Leica, Wetzlar, Germany) and stored floating in a cryoprotective solution (25% ethylene glycol and 25% glycerol in PBS (Sigma-Aldrich, Darmstadt, Germany) at 4 °C. For IHC, five sections (30 µm) from each mouse were washed and blocked for 1 h in 5% horse serum in PBS with 0.5% Triton X-100 (Sigma-Aldrich, Darmstadt, Germany) at room temperature (RT). Primary antibodies were diluted in 3% horse serum (Jackson ImmunoResearch, West Grove, Pennsylvania, USA) in PBS/0.3% Triton X-100 and sections incubated for 48 h at 4 °C, followed by washing and incubation with the respective secondary antibodies for 2 h at RT. For nuclear counterstaining, 4′,6-diamidino-2-phenylindole (DAPI, Sigma, Missouri, United States) was used. The sections were then mounted on SuperFrostPlus slides (ThermoFisher, Darmstadt, Germany) with Aqua-Polymount (Polysciences, Inc. Warrington, PA, USA). For the direct visualization of tdTomato, sections were only counterstained with nuclear stain (DAPI) and investigated by confocal microscope (Leica TCS SP5-II; Mannheim, Germany). The primary antibodies applied were anti-NeuN (1: 1000, mouse; Millipore, Darmstadt, Germany), and anti-Ctip2 (1: 500, guinea pig; SYSY, Göttingen, Germany). The secondary antibodies used were Alexa488 anti-guinea pig (1: 500, Invitrogen, Darmstadt, Germany), and Alexa647 anti-mouse (1: 500; ThermoFisher, Darmstadt, Germany). For the quantification of NeuN+tdTomato+ co-labelled cells, sequential coronal sections from the dorsal part of hippocampus were taken (coordinates from Bregma: −1.34 to −2.54 mm posterior). Stained sections (30 µm) were imaged using Leica TCS SP5 (Mannheim, Germany), equipped with a 20× objective (NA = 0.70). Hypoxic neurons (Ctip2+tdTomato+) from CA1, CA3, and the dentate gyrus were counted manually by Fiji software (https://imagej.net/Fiji, accessed on 18 February 2021). Representative images of bilateral hippocampi from five sections per mouse were processed/analyzed using Imaris 9.1.0 (www.bitplane.com, accessed on 18 February 2021). 

### 3.11. Light-Sheet Microscopy (LSM)

Whole mount tissue staining and clearing:

To visualize the tdTomato+ cells in the entire brain, we performed LSM in combination with whole mount immune labelling and tissue clearing. Following transcardial perfusion as described above, the brains were removed, post-fixed in 4% PFA overnight and stored in PBS. The brain hemispheres were processed for immune-labelling and tissue clearing following a slightly modified iDISCO protocol [38]. The samples were dehydrated with a methanol/PBS series (50%, 80%, 100%, and 2 × 100%, 1 h, RT) followed by overnight bleaching and permeabilization in a mixture of 5% H_2_O_2_/20% dimethyl sulfoxide (DMSO) (Sigma-Aldrich, Darmstadt, Germany) in methanol at 4 °C. Samples were retrieved and washed further with methanol at 4 °C for 30 min and −20 °C for 3 h prior to incubation in 20% DMSO in methanol at RT for 2 h. Samples were then rehydrated using a descending methanol/PBS series (80%, 50%, PBS, 1 h each, RT) and further washed with in PBS/0.2% TritonX-100 for 2 h. The samples were then incubated overnight in 0.2% TritonX-100, 20% DMSO, and 0.3 M glycine in PBS at 37 °C and blocked using PBS containing 6% goat serum, 10% DMSO and 0.2% Triton-X100 for two days at 37 °C. The samples were then washed twice in PBS containing 0.2% Tween(r)20 and 10 µg/mL heparin (PTwH) at RT for 1 h and incubated with primary antibody solution (anti-RFP; Rockland # 600-401-379, Pennsylvania, USA; 1: 250 in PTwH/5%DMSO/3% goat serum) for 14 days at 37 °C. After several washes during the day and an additional overnight wash in PTwH, the samples were incubated with secondary antibody solution (goat anti-rabbit Alexa555; ThermoFisher Scientific Darmstadt, Germany, # A-21428; 1: 500 in PTwH/3% goat serum) for seven days at 37 °C. Prior to clearing, the samples were again washed in PTwH (several solution changes during the day) followed by an additional overnight wash. Tissues were dehydrated using an ascending series of methanol/PBS (20, 40, 60, 80, and 2 × 100% 1 h, RT) followed by overnight incubation in a mixture of 33% dichloromethan (DCM) and 66% methanol at RT. Samples were further delipidated by incubation in 100% DCM for 40 min and transferred to pure ethyl cinnamate (Eci; Sigma Aldrich #112372, Darmstadt, Germany) as the clearing agent. The tissues became transparent after 30 min in Eci and were stored at RT until imaging.

LSM and 3D analysis/visualisation:

LSM was performed using a LaVision Ultramicroscope II equipped with a 2× objective, corrected dipping cap and zoom body. Samples were mounted onto the sample holder with the medial surface of the brain hemisphere facing down in order to acquire sagittal images. The holder was placed into the imaging chamber filled with Eci. The images were acquired in mosaic acquisition mode with the following specifications: 5 µm sheet thickness; 20% sheet width; 2x zoom; 4 × 5 tiling; 4 µm z-step size; dual site sheet illumination; 50 ms camera exposure time; 1000 px × 1600 px field of view. Red fluorescence was recorded using 561 nm laser excitation (5 to 10%) and 585/40 emission filters. Images were loaded into Vision4D 3.0 (Arivis) and stitched using the tile sorter setup. Hippocampus and cortex regions of interests (ROIs) where manually annotated according to the sagittal Allen mouse brain atlas [39]. For this, hippocampus and cortex ROIs were traced manually in a few planes in 2D from which the 3D ROI was extrapolated automatically. Cortex and hippocampus annotations were then cropped with a medial cut-off of approximately 0.4 mm and a lateral cut-off of approximately 4.4 mm (corresponding to the lateral end of the hippocampal formation). Cortex ROIs spanned the dorsal parts of the cortex as defined by anatomical landmarks. Next, tdTomato+ cells per ROI were identified using the blob finder algorithm in Vision4D. Noise was removed by deleting objects with voxel sizes < 10 from the object table. Objects were then critically reviewed, and any additional noise was manually removed from the dataset. The number of tdTomato+ cells per ROI was extracted from the object table and plotted using GraphPad (Prism). For visualisation purposes, a 1300 × 1400 × 100 px field of view spanning the hippocampus and overlaying the cortex was cut out from the original datasets and visualized in the 3D maximal intensity modus. Whole-brain and hippocampal 3D videos as well as ‘flythrough’ stacks were rendered in Vision4D.

### 3.12. scRNA-seq

Single-cell RNA sequencing (scRNA-seq) data was obtained and processed as described in detail earlier [7] (GSE162079). For the purpose of the current paper, only excitatory neuron populations known to express *CaMKIIα* were included in the analyses. All analyses were performed using R v4.0.0 [40], except for the MiloR package, which was run in R v4.0.3. Differentially expressed genes between hypoxia and normoxia were identified using the FindMarkers() function with default settings (two tailed correlation-adjusted Mann-Whitney U test, |log-fold change| > 0.25) in Seurat v3.1.5 [41] (please note that the empty brackets indicate the use of respective functions with its default settings). The regional assignment of neuronal clusters was performed using the following markers: *Mpped1* for CA1, *Mndal* for CA3, and *Prox1* for DG. Mossy fiber cells were characterized by the strong expression of *Calb2* [42], while the Glut3 cluster was showing a high expression of *Tshz2*, possibly indicating subicular origin [43]. Glut4 was high in the expression of *Tbr1*, *Dcx*, and *Tle4*, indicating an immature cell state [6]. In order to mitigate power-related differences in the number of detectable differentially expressed genes, clusters were down-sampled to the smallest cluster included in the comparison, i.e., CA3 with *n* = 2296 (Figure 6c, left panel). For easier visualization, a small number of cells located far below on the UMAP2 axis were shifted upwards. The heatmap was created with the ComplexHeatmap package [44]. The Venn diagrams were created with with VennDiagram [45]. The numerical changes under hypoxia versus normoxia were investigated using the MiloR v0.1.0 [20] package. The graph was built using buildGraph(*k* = 30, *d* = 30) and neighbourhoods were defined with the same paramaters by makeNhoods() and setting the proportion of randomly sampled graph vertices to prop = 0.2. Distances between the nearest neighbours were calculated with calcNhoodDistance(*d* = 30). Cells per neighbourhood were quanitfied with countCells(). Differential abundance testing was performed with testNhoods(design= ~experimental_group) and the results were visualized after calling buildNhoodGraph() with plotNhoodGraphDA(). On https://github.com/AgnesSteixner/Butt_et_al_Camk2a, accessed on 18 February 2021, analysis scripts are available. Bonferroni-adjusted (Seurat) or false discovery rate-adjusted (MiloR) *p*-values were reported.

### 3.13. Statistical Analysis

Data obtained for all quantifications were analyzed by Graph Pad Prism 8 (GraphPad Software, Inc. San Diego, CA, USA). The statistical significance was calculated by using an unpaired Welch’s *t*-test to compare two groups. A *p* value < 0.05 was considered statistically significant. All bar graphs show means and error bars represent the standard error of mean (SEM).

## Figures and Tables

**Figure 1 ijms-22-03164-f001:**
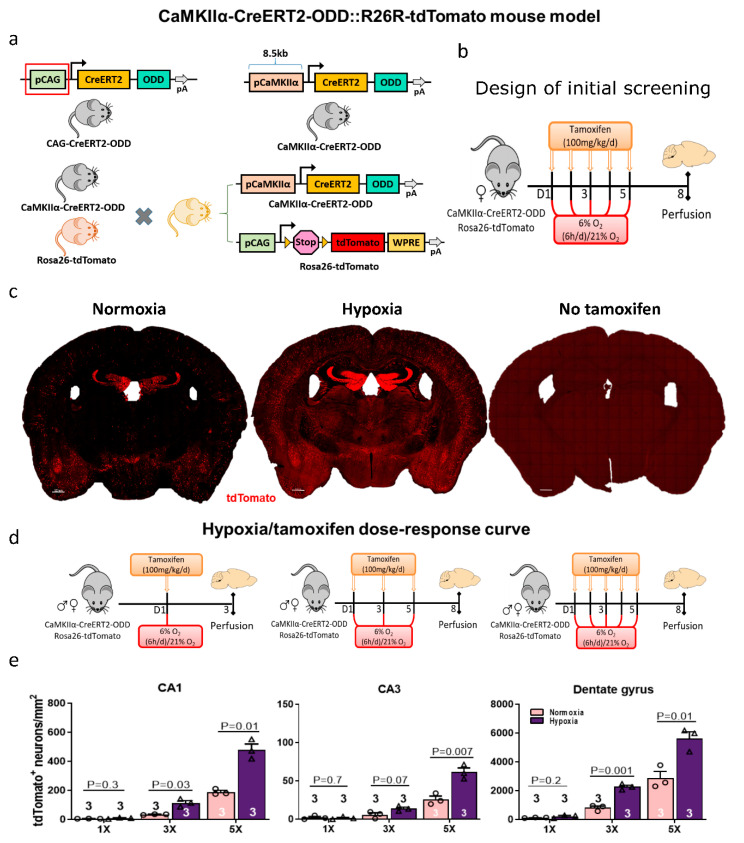
Generation and first characterization of excitatory neuron-specific CaMKIIα-CreERT2-ODD::R26R-tdTomato hypoxia reporter mice. (**a**) Schematic representation of the CAG-CreERT2-ODD construct where the CAG promoter was excised using SpeI and EcoRI restriction enzymes to obtain the vector backbone (CreERT2-ODD-pA). Subsequently, the excitatory neuronal promotor CaMKIIα of 8.5 kb was inserted into CreERT2-ODD-pA to generate the CaMKIIα-CreERT2-ODD-pA vector. Next, the linearized vector was used for the generation of CaMKIIα-CreERT2-ODD expressing transgenic mice. These were crossed with CAG-Rosa26-tdTomato reporter mice. Permanent labelling of hypoxic neurons is achieved via Hif-1α oxygen-dependent degradation domain (ODD) stabilization upon tamoxifen injection. (**b**) Experimental design: For an initial screening, CaMKIIα-CreERT2-ODD::R26R-tdTomato hypoxia reporter mice received tamoxifen five times over five days, each time followed by exogenous hypoxia (6% O_2_ for 6 h) versus normoxia (21% O_2_), and mice were sacrificed on day eight. (**c**) Representative coronal images of five-times tamoxifen-injected CaMKIIα-CreERT2-ODD::R26R-tdTomato mice under normoxic or hypoxic conditions. The corn oil control (‘no tamoxifen’) shows only very few tdTomato+ (red) neurons, i.e., minimal tamoxifen-independent Cre-activity. Scale bar represents 500 µm. (**d**) Experimental scheme of the hypoxia/tamoxifen dose response curve in CaMKIIα-CreERT2-ODD::R26R-tdTomato mice. (**e**) Quantification of tdTomato+Ctip2+ double-positive neurons in CA1, CA3 and dentate gyrus from eight-week-old mice of equally distributed mixed gender. Unpaired Student’s *t*-test (two-tailed, Welch’s corrections) was used for statistical analysis between groups; n numbers given in graphs; error bars indicate standard error of mean (SEM).

**Figure 2 ijms-22-03164-f002:**
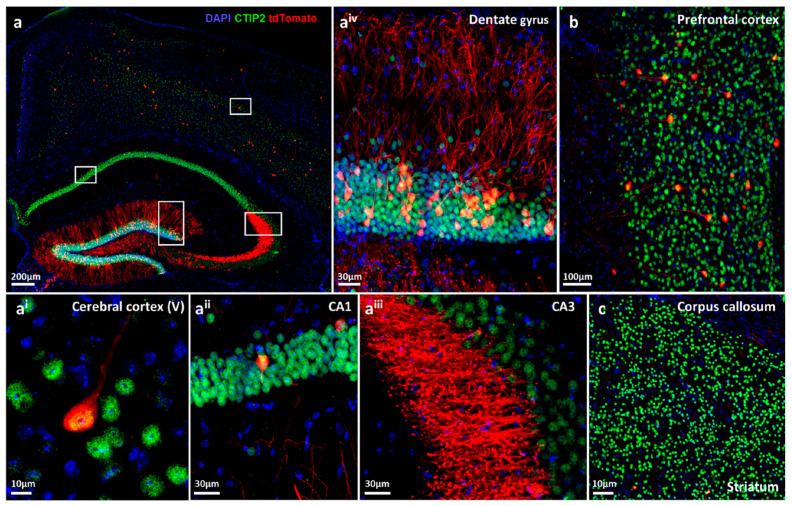
Immunohistochemical characterization of functional hypoxic pyramidal neurons in CaMKIIα-CreERT2-ODD::R26R-tdTomato reporter mice. (**a**) Overview image of an eight-week-old female mouse after exposure to complex running wheel (CRW) showing distinct labelling of neurons with tdTomato fluorescent protein (red), co-immunostained with the neuronal marker Ctip2 (green) and DAPI (blue) as nuclear counterstain. White frames denote magnifications given in **a^i^**–**a^iv^**. (**a****^i^**) Image of cortical layer V showing a neuron with tdTomato+ and Ctip2+ co-localized signal. (**a^i^****^i^**) Some cornu ammonis (CA1) pyramidal neurons are tdTomato positive. (**a^ii^****^i^**) In CA3, just a few neuronal cell bodies display the tdTomato signal; however, strong staining is observed in stratum lucidum, rendering neuronal morphological details such as dendritic spines and synapses easily distinguishable at high resolution. (**a^iv^**) Close-up image of the superior plate of the dentate gyrus, demonstrating densely tdTomato-expressing granular neurons and remarkable labelling of neutropil. (**b**) A subset of neurons in prefrontal cortex and their processes are also tdTomato labelled. (**c**) CaMKIIα expressing tdTomato+ neurons are sparsely present in the striatum. Scale bars represent 200 µm (**a**), 10 µm (**a^i^, c**), 30 µm (**a^ii^**, **a^iii^** and **a^iv^**), and 100 µm (**b**).

**Figure 3 ijms-22-03164-f003:**
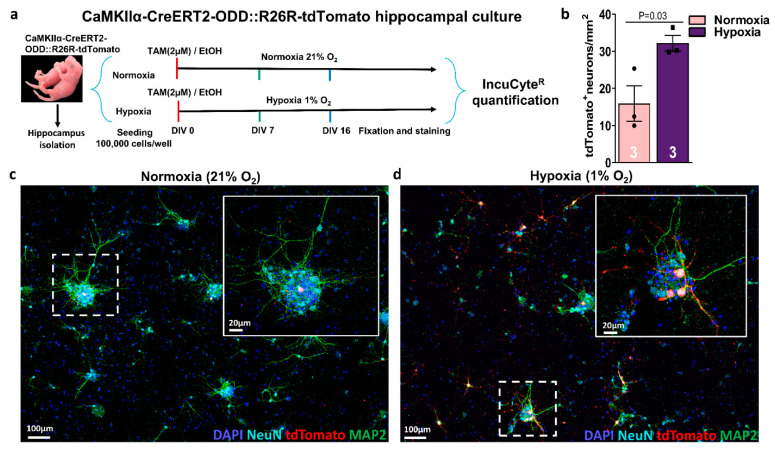
Maturing hippocampal neurons from CaMKIIα-CreERT2-ODD::R26R-tdTomato mice respond to hypoxia also in vitro. (**a**) Experimental design: Isolation and culture of hippocampal neurons from E17 pups. (Z)-4-hydroxytamoxifen (2 µM) or solvent control (final EtOH concentration always <0.016%) are added to neuron cultures on day in vitro (DIV)0, followed by incubation in the IncuCyte^R^ under either normoxia (21% O_2_) or hypoxia (1% O_2_) for seven or 16 days, fixation, staining, and quantification. (**b**) Quantification of hypoxic neurons, as visualized by tdTomato+ label, reveals an increase under hypoxia at DIV16. (**c**,**d**) Representative images of hippocampal neurons stained with neuronal markers NeuN (light blue), MAP2 (green) demonstrate co-localization with tdTomato (red), as shown strongly in hypoxic and less prominent in normoxic conditions. DAPI (blue) was used as a nuclear counterstain. Scale bars represent 100 µm in overview and 20 µm in magnified images. Unpaired Student’s *t*-test (one-tailed, Welch’s corrections) was used for statistical analysis between conditions; n numbers given in graphs; error bars indicate standard error of mean (SEM).

**Figure 4 ijms-22-03164-f004:**
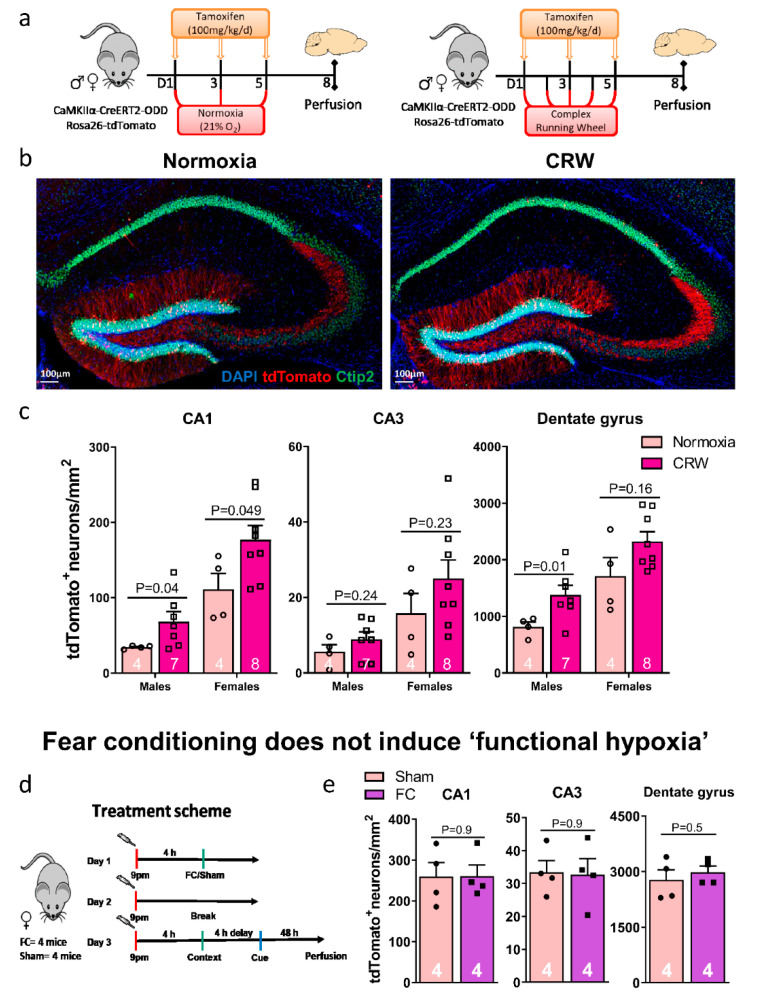
Complex motor-cognitive challenge as inducer of ‘functional hypoxia’ in the hippocampus. (**a**) Experimental design: mice with no prior training were given free access to complex running wheels (CRW) for five complete days and received tamoxifen injections every other day (three in total). Normoxia mice obtained the same treatment but were kept in standard cages. All mice were perfused on day eight and brains collected for histology. (**b**) Representative hippocampus images from eight-week-old female CRW-exposed and normoxia mice; scale bar 100 µm. (**c**) Quantifications of tdTomato+/Ctip2+ neurons in CA1, CA3, and dentate gyrus revealed an increase upon exposure to CRW in both genders, with normoxia and CRW values considerably more pronounced in females. (**d**) Experimental design of the fear conditioning (FC) pilot experiment (24-week-old females). (**e**) Quantification of tdT+/Ctip2+ neurons in hippocampal CA1, CA3, and dentate gyrus regions of sham and FC mice indicate no noteworthy difference. This is likely due to the FC stimulus being too short-lived to evoke functional hypoxia. Unpaired Student’s *t*-test (two-tailed, Welch’s corrections); n numbers given in graphs; error bars indicate standard error of mean (SEM).

**Figure 5 ijms-22-03164-f005:**
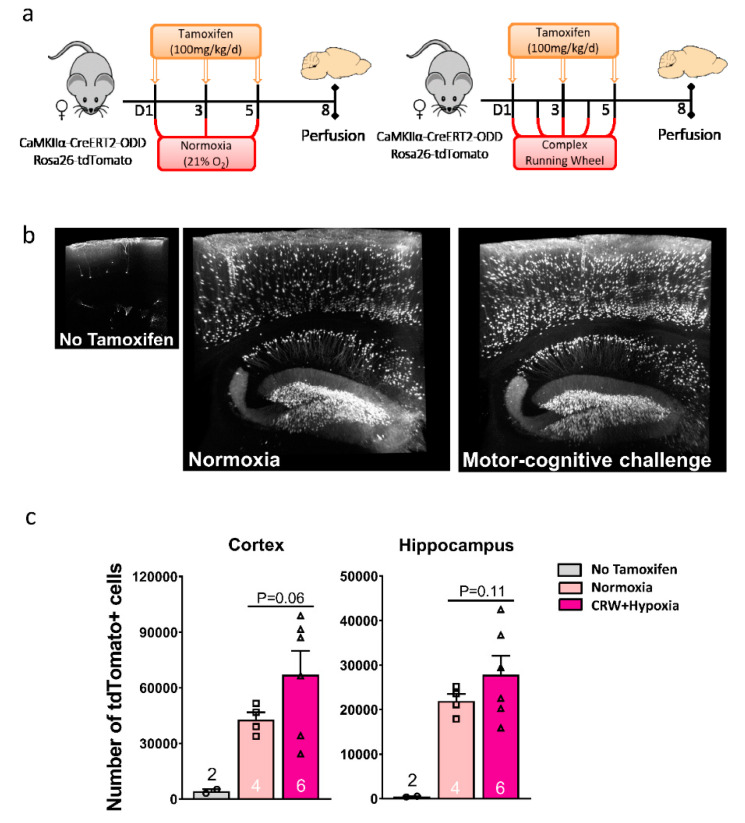
Light-sheet microscopy (LSM) enables 3D presentation of hypoxic neurons in CaMKIIα-CreERT2-ODD::R26R reporter mice. (**a**) Experimental design. (**b**) Illustrative 3D images rendered in maximal intensity modus demonstrate tdTomato+ hypoxic neurons in hippocampus and cortex of female normoxia versus CRW mice; the small image on the left displays a ‘no tamoxifen’ control brain with very few scattered tdTomato+ neurons. Scale: 1.96 × 2.28 × 0.4 mm. (**c**) LSM quantification of tdTomato+ cells in cortex and hippocampus of eight-week-old female mice show a clear tendency of an equally strong increase under CRW and hypoxia (groups thus pooled) as compared to normoxia, with the ‘no tamoxifen’ condition being negligible. Unpaired Student’s *t*-test (two-tailed, Welch’s corrections); n numbers given in graphs; error bars indicate standard error of mean (SEM).

**Figure 6 ijms-22-03164-f006:**
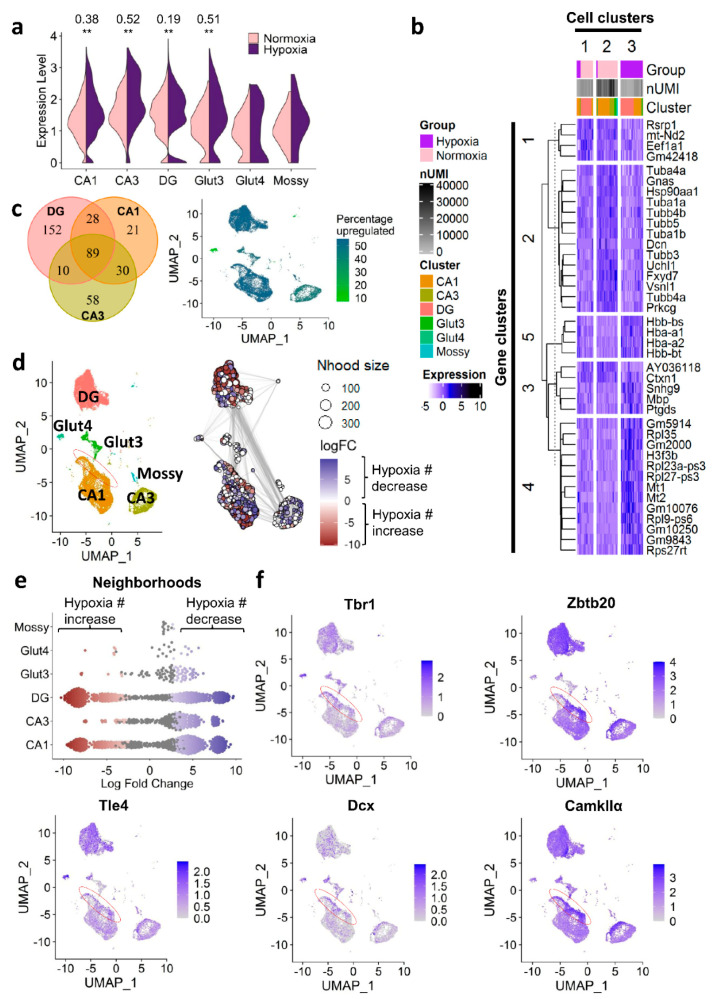
Response of hippocampal excitatory neurons to hypoxia evaluated by scRNA-seq. (**a**) Upregulation of *CaMKIIα* under hypoxia (five days, 6% O_2_ for 6 h/day) in excitatory neurons of the hippocampus. (**b**) Heatmap of top 20 upregulated and top 20 downregulated genes under hypoxia. Columns represent cells, rows represent genes. Scaled normalized expression shown. Cells and genes were clustered using the k-means algorithm. Number of unique molecular identifiers per cell: nUMI. (**c**) Left panel: Venn diagram depicting number of shared and uniquely regulated genes under hypoxia amongst excitatory neurons of different hippocampal regions (CA1, CA3, dentate gyrus). All clusters were randomly down-sampled to *n* = 2296 cells for better comparability. Right panel: Percentage of upregulated genes amongst all hypoxia-regulated genes; |log-fold change| > 0.25, *p*-adj < 0.05, correlation-adjusted Wilcoxon test. (**d**) Differential cell abundance testing with MiloR [20]. Red (= negative logFC) represents a numerical increase upon hypoxia in the respective neighborhood, blue (= positive logFC) indicates a decrease. Region marked in red specifies region of immature and intermediately mature neurons of CA1 increased in number under hypoxia. logFC: log-fold change, Nhood size: neighborhood size. (**e**) Beeswarm plot showing the numerical increases and decreases under hypoxia in the respective neighborhoods of each cell cluster. Red (=negative logFC) indicates an increase under hypoxia, blue (= positive logFC) a decrease. (**f**) Expression of immature and intermediately mature excitatory neuron markers. Red oval indicates again the interesting region denoted in (**d**) with increase in cell abundance upon hypoxia.

## Data Availability

Analysis scripts for scRNA-seq analysis: https://github.com/AgnesSteixner/Butt_et_al_Camk2a. Raw and processed scRNA-seq data are publicly available on GEO via accession code GSE162079.

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
