# Peer review of "CaMKIIα Expressing Neurons to Report Activity-Related Endogenous Hypoxia upon Motor-Cognitive Challenge"

_ijms, 2021, doi:10.3390/ijms22063164_

Round 1

Reviewer 1 Report

The authors documented that the task-associated activity triggers neuronal functional hypoxia as local and brain-wide reaction mediating adaptive neuroplasticity. The manuscript is well organized and clearly documented. I have one comment to improve.

  1. Although the authors concluded that the hypoxia-induced genes like EPO drive neuronal differentiation, brain maturation and improved performance, However, the relationships between EPO induction and brain maturation are largely unknown. The authors should define the mechanism underlying EPO-induced neural maturation in the manuscript.

Author Response

POINT-BY-POINT RESPONSE to the REVIEWERS’ COMMENTS
(Reviewers comments in Times New Roman 12; our response in Arial 12 – blue, with changes in the manuscript announced in red.)

Reviewer 1:

The authors documented that the task-associated activity triggers neuronal functional hypoxia as local and brain-wide reaction mediating adaptive neuroplasticity. The manuscript is well organized and clearly documented. I have one comment to improve.

We thank the reviewer for the time invested into reviewing our manuscript and for the positive feedback!

Although the authors concluded that the hypoxia-induced genes like EPO drive neuronal differentiation, brain maturation and improved performance, However, the relationships between EPO induction and brain maturation are largely unknown. The authors should define the mechanism underlying EPO-induced neural maturation in the manuscript.

We have now addressed this point and added the following explanation in the manuscript, page 12, including two more references.

These findings highlight that EPO is not only a crucial mediator of neurogenesis during embryonic life and brain development [30, 31], but also pivotal for adult hippocampal neurodifferentiation and neuroplasticity on demand. The neuroprotective and antiapototic effects of EPO, promoting neuronal survival, together with its neurotrophic properties, certainly contribute to the observed enhanced differentiation of local silent precursors and to their undisturbed maturation. From the discovery of this regulatory EPO circle, driving challenge-induced brain maturation,...
Constanthin, P. E.; Contestabile, A.; Petrenko, V.; Quairiaux, C.; Salmon, P.; Hüppi, P. S.; Kiss, J. Z., Endogenous erythropoietin signaling regulates migration and laminar positioning of upper-layer neurons in the developing neocortex. Development 2020, 147, (19), dev190249.
Noguchi, C. T.; Asavaritikrai, P.; Teng, R.; Jia, Y., Role of erythropoietin in the brain. Critical reviews in oncology/hematology 2007, 64, (2), 159-71.

Reviewer 2 Report

The MS presents new look at the functional hypoxia as a regulator of neuroplasticity and maturation of the excitatory neurons in the hippocampus in Hif dependent manner. The authors postulate that CA1 pyramidal neurons are specifically regulated by hypoxia, but according to Fig. 2 in the dentate gyrus there are many more dTomato-expressing granular cells than in CA1. It seems that all types of glutamatergic neurons (pyramidal and granular) in the hippocampus specifically respond to hypoxia. In vitro data indicate that only immature/mature postmitotic cells respond to hypoxia suggesting that an increase in expression of Tbr1, Zbtb20, Tle4, and Dcx upon hypoxia in CA1 may reflect activation of silent postmitotic progenitors or it probably could be also associated with the activation of glutamatergic transmission, as it was shown that Tbr1 is activated in an activity-dependent manner via NMDAR (Chuang et al., 2014).

Minor comments:

The authors divided the hippocampal cells for neuronal clusters, but mentioned only markers for CA1 (Mpped1), CA3 (Mndal) and DG (Prox1). Please clarify how the mossy, Glut3 and Glut4 clusters were identified.

Please clarify empty brackets - lines 588, 599, 604.

Author Response

Reviewer 2:

The MS presents new look at the functional hypoxia as a regulator of neuroplasticity and maturation of the excitatory neurons in the hippocampus in Hif dependent manner. The authors postulate that CA1 pyramidal neurons are specifically regulated by hypoxia, but according to Fig. 2 in the dentate gyrus there are many more tdTomato-expressing granular cells than in CA1. It seems that all types of glutamatergic neurons (pyramidal and granular) in the hippocampus specifically respond to hypoxia. In vitro data indicate that only immature/mature postmitotic cells respond to hypoxia suggesting that an increase in expression of Tbr1, Zbtb20, Tle4, and Dcx upon hypoxia in CA1 may reflect activation of silent postmitotic progenitors or it probably could be also
3
associated with the activation of glutamatergic transmission, as it was shown that Tbr1 is activated in an activity-dependent manner via NMDAR (Chuang et al., 2014).

We thank the reviewer for reading our manuscript and the encouraging feedback.
We agree with the reviewer, that all types of glutmatergic neurons are undergoing hypoxia (see differential expression results in Figure 6b and c). However, high number of tdTomato-expressing granular cells are observed in the dentate gyrus due to differences in the level of CaMKIIα expression and cell density. These differences might indicate a particular and distinct response of the dentate gyrus as compared to the CA region as also discussed in lines 288 to 296. We agree that hypoxia could be activating silent postmitotic progenitors in the CA1 and have added this possibility now in the manuscript discussion (line 365).

Minor comments:

The authors divided the hippocampal cells for neuronal clusters, but mentioned only markers for CA1 (Mpped1), CA3 (Mndal) and DG (Prox1). Please clarify how the mossy, Glut3 and Glut4 clusters were identified.

Mossy fiber cells were characterized by the strong expression of Calb2 (see also Cembrowski et al 2016), while the Glut3 cluster was showing high expression of Tshz2 (possibly indicating subicular origin, see Harris et al., 2018). Glut4 was high in expression of Tbr1, Dcx and Tle4, indicating an immature cell state (Wakhloo et al., 2020) as mentioned in line 303/304. See added text page 16.
Cembrowski, M. S.; Wang, L.; Sugino, K.; Shields, B. C.; Spruston, N., Hipposeq: a comprehensive RNA-seq database of gene expression in hippocampal principal neurons. Elife 2016, 5, e14997.
Harris, K. D.; Hochgerner, H.; Skene, N. G.; Magno, L.; Katona, L.; Bengtsson Gonzales, C.; Somogyi, P.; Kessaris, N.; Linnarsson, S.; Hjerling-Leffler, J., Classes and continua of hippocampal CA1 inhibitory neurons revealed by single-cell transcriptomics. PLOS Biology 2018, 16, (6), e2006387.
Wakhloo, D.; Scharkowski, F.; Curto, Y.; Butt, U. J.; Bansal, V.; Steixner-Kumar, A. A.; Wüstefeld, L.; Rajput, A.; Arinrad, S.; Zillmann, M. R., Functional hypoxia drives neuroplasticity and neurogenesis via brain erythropoietin. Nature communications 2020, 11, (1), 1-12.
Please clarify empty brackets - lines 588, 599, 604.
Empty brackets indicate the use of the respective functions with default settings. This is now added at the respective position of first presentation of () in the text page 16